# What is the coverage of retina screening services for people with diabetes? Protocol for a systematic review and meta-analysis

Nimisha Chabba [1] Pushkar Raj Silwal [1] Covadonga Bascaran,[2] Ian McCormick [2] Lucy Goodman,[1] Iris Gordon,[2] Matthew J Burton,[2,3] Stuart Keel,[4] Jennifer Evans,[2] Jacqueline Ramke [1,2]

[1]School of Optometry and Vision Science, University of Auckland, Auckland, New Zealand
[2]International Centre for Eye Health, London School of Hygiene and Tropical Medicine, London, UK
[3]National Institute for Health Research Biomedical Research Centre for Ophthalmology, Moorfields Eye Hospital NHS Foundation Trust and UCL Institute of Ophthalmology, London, UK
[4]Department of Noncommunicable Disease, World Health Organization, Geneva, Switzerland

**Correspondence to**
Nimisha Chabba;
nimisha.chabba@auckland.ac.nz

## ABSTRACT

**Introduction** Diabetic retinopathy is a leading cause of vision impairment globally. Vision loss from diabetic retinopathy can generally be prevented by early detection and timely treatment. The WHO included a measure of service access for diabetic retinopathy as a core indicator in the Eye Care Indicator Menu launched in 2022: *retina screening coverage for people with diabetes*. The aim of this review is to provide a comprehensive global and regional summary of the available information on retina screening coverage for people with diabetes.

**Methods and analysis** A search will be conducted in five databases without language restrictions for studies from any country reporting retina screening coverage for adults with any type of diabetes at the national or subnational level using data collected since 1 January 2000 until the search date. We will also seek reports and coverage statistics from government websites of all WHO member states. Two investigators will independently screen studies, extract relevant data and assess risk of bias of included studies. The results of the review will be reported using the Preferred Reporting Items for Systematic Review and Meta-Analysis guideline. We will summarise the range of coverage definitions reported across included studies and present the median retina screening coverage in WHO regions and by World Bank country income level. Depending on the availability of data, we will conduct meta-analysis to assess disparities in retina screening coverage for people with diabetes by factors in the PROGRESS framework (Place of residence, Race/ethnicity/culture/language, Occupation, Gender/sex, Religion, Education, Socioeconomic status and Social capital).

**Ethics and dissemination** This review will only include published data thus no ethical approval will be sought. The findings of this review will be published in a peer-reviewed journal and presented at relevant conferences.

**Protocol registration number** OSF registration 17/10/2023: https://osf.io/k5p69.

## INTRODUCTION

Diabetic retinopathy is a leading cause of vision impairment globally, and the only one of the five leading causes that increased in prevalence in the last 30 years.[1] Most of the

### STRENGTHS AND LIMITATIONS OF THIS STUDY

⇒ A strength of this review is the comprehensive search, designed by an experienced information specialist, and without any language or geographical restrictions, across several databases for reports with data collected since 1 January 2000 until the search date.
⇒ In addition, an extensive and systematic search of government websites (grey literature) will be undertaken and field experts will be requested to nominate additional potentially relevant reports.
⇒ A further strength is that study selection, data extraction and quality assessment will be conducted by two investigators independently.
⇒ A potential limitation is that anticipated heterogeneity in the definitions of the outcome measure, reporting period and method of data collection may preclude quantitative synthesis of results.

vision impairment caused by diabetic retinopathy is avoidable through early detection and timely treatment.[2 3] However, vision that has already been lost due to diabetic retinopathy cannot be restored.[4] Many countries have retina screening programmes for diabetic retinopathy (also called diabetic eye screening or diabetic retinopathy screening) that are administered at a national-level, regional-level or facility-level and aim to identify all patients with sight-threatening retinopathy, with subsequent treatment by the same service or via referral to (generally co-located) treatment services.[5]

Diabetic retinopathy screening programmes will only be successful in reducing vision loss if most people with diabetes are routinely screened and referred for treatment when indicated. The recommended screening interval varies between countries (eg, 1 year in Kenya, 1–2 years in England and 2–3 years in New Zealand).[5 6] Regardless of

the recommended period, available evidence confirms that attendance at retinal screening is suboptimal, with services being less accessible for some population groups.[7 8] People who experience delayed or reduced access to screening will be less likely to be offered timely initiation of treatment should they require it.[9]

A measure for diabetic retinopathy service access: *retina screening coverage for people with diabetes* was recently proposed by WHO as a core indicator for eye health.[10] This indicator uses routinely collected health facility data or population-based surveys to measure the percentage of people with known/diagnosed diabetes who undertake a retina examination at the interval recommended and defined in nationally adopted guidelines. In countries without a recommended screening interval, WHO recommends screening every 2 years. The way that studies define or report coverage varies. For example, while some studies report coverage at a specific point in time from cross-sectional facility-based data[11] or self-reported service use in survey data,[12] others report the proportion of the enrolled diabetic population who attend all expected screening events within a longer period of time (eg, 5 years of follow-up[13 14]).

Measuring retina screening coverage is essential to help monitor progress towards the prevention of sight-threatening diabetic retinopathy.[15] WHO recently included this indicator in the District Health Information Systems, version 2 to support the integration and systematic collection of eye care data into facility level reporting systems.[16] To date, retina screening coverage for people with diabetes has not been summarised systematically at the global or regional level.

The aim of this review is to provide a comprehensive global and regional summary of the available information on retina screening coverage for people with diabetes.

## Objectives

The objectives of this review are to:

1. Summarise the different ways retina screening coverage for people with diabetes has been defined in published reports.
2. Summarise retina screening coverage for people with diabetes by World Bank country income level and WHO regions.
3. Compare disparities in retina screening coverage for people with diabetes by factors in the PROGRESS framework (ie, between subgroups within any of: Place of residence, Race/ethnicity/culture/language, Occupation, Gender/sex, Religion, Education, Socioeconomic status and Social capital).[17]

## METHODS AND ANALYSIS

This protocol is reported in accordance with the Preferred Reporting Items for Systematic Review and Meta-Analysis Protocols (PRISMA-P) checklist.[18 19] The completed PRISMA-P checklist is presented in online supplemental annex 1. The protocol has been registered with the Open Science Framework registry and can be viewed online https://osf.io/k5p69.

## Eligibility criteria
### Population and context
We will include reports from any country reporting retina screening coverage for adults for any type of diabetes. There will be no restrictions on patient characteristics such as sex, ethnicity, duration of diabetes or location. We will exclude studies reporting outcomes only in children (aged 18 years or below) due to the higher likelihood of people <18 years having type 1 diabetes, with a delayed first screening recommendation. We will exclude studies reporting outcomes only for people with pre-diabetes, and studies reporting outcomes only for people with gestational diabetes because it is mainly a temporary type of diabetes with different screening recommendations.

### Type of studies
1. Type of report: we will include published (peer reviewed) literature where the full report is published, that is, we will exclude reports with only an abstract available, as well as editorials, protocols and reports of pilot studies. We will also exclude systematic reviews or other evidence syntheses but will examine the reference list of relevant reviews to identify potentially relevant primary studies. We will include grey literature from governments that meet our other inclusion criteria.
2. Source of data: we will include data derived from any of
   i. Population-based research studies.
   ii. Facility-based data collected routinely or de novo.
   iii. Service activity or payment data from Ministries of Health or health insurance databases.

   These sources of data may be reported at the national or subnational (state/province/district) level. We will exclude studies reporting data where the sampling is not representative of the national or state/province/district level.
3. Study design: we will include cross-sectional studies, cohort studies, randomised controlled trials (for the control arm(s)) and non-randomised intervention studies (for the control arm(s)). We will exclude studies only reporting modelled data.
4. Other: due to the changing nature of the health system over time, we will limit our review to data collected since 1 January 2000. We will exclude studies reporting data exclusively during the peak of the COVID-19 pandemic (2020 and 2021) due to the range of responses and restrictions across the world that reduced health service delivery. If an included study reports estimates from both COVID-19 and non-COVID-19 years, we will include data from the non-COVID-19 period. Our search will have no language restrictions and every effort will be made to translate reports in languages other than English.

## Outcome

We will include studies that report a measure of retina screening coverage that includes a numerator and denominator value at a national, or state/province/district level, regardless of the screening interval used (eg, annual, biennial). We will also include studies that report the number of people with diabetes who attend all of their expected retina screening appointments during a defined follow-up period. The screening could involve a retinal examination by a relevant health practitioner or where an attempt is made to image the retina of a patient by digital photography to determine the presence (and grading) of diabetic retinopathy by a human grader and/or by artificial intelligence. We will exclude studies reporting a coverage proportion without also reporting at least one of either the numerator or denominator and a defined screening interval or time period. We will also exclude studies only reporting attendance at appointments subsequent to retinal screening (eg, referral to an eye clinic for further assessment).

When a potentially eligible study only reports coverage disaggregated by population groups and not overall, we will cross-check our included studies to determine if the overall coverage proportion is available in a separate publication; in the event it is not, we will contact study authors to request the overall coverage. If the overall coverage can be provided, we will include the study. In the event the overall coverage cannot be provided, we will include the study only if the population groups for which the data are disaggregated are relevant to the PROGRESS framework, and this study will be considered for analysis of objectives 1 and 3 but not objective 2. For example, we will include a study disaggregating coverage for immigrants and non-immigrants, but not a study disaggregating coverage for people with different levels of vision difficulty.

### Information sources and search strategy

Using a search strategy developed by an experienced information specialist, we will search MEDLINE (Ovid), Cochrane Central Register of Controlled Trials (CENTRAL), Embase, Global Health (Ovid), Emerging Sources Citation Index (ESCI), Science Citation Index Expanded (SCI-EXPANDED) and Social Sciences Citation Index (SSCI) on Web of Science (Clarivate) for reports using data collected since 1 January 2000. Our search strategy is included in online supplemental annex 2. We will download and de-duplicate the results in EndNote, and then export the results into Covidence (Veritas Health Innovation, Melbourne, Australia; available at www.covidence.org) for screening. We will examine reference lists of all included reports to identify further potentially relevant reports.

We will search for grey literature in the form of government reports of retina screening coverage at the national or subnational level. Grey literature reports will be searched for by one investigator (NC); another investigator (LG) will repeat the searches for 20 countries to ensure the consistency of relevant results. Government websites, including that of national and provincial government health departments, of the 194 WHO member states and eight associate countries and territories (American Samoa, Bermuda, French Polynesia, Greenland, Hong Kong, Palestine, Puerto Rico and Tokelau) will be identified and searched for relevant reports and coverage statistics. The process for systematically collecting government reports will be based on the grey literature search framework outlined by Godin *et al*.[20] First, to identify the relevant government websites, a Google search will be conducted using the search strategy in online supplemental annex 3. The first ten pages of each search (representing 100 results) will be reviewed, using the title and short text underneath. The relevant websites' name/organisation, country and URL will be entered into an Excel spreadsheet. Then, each of these websites will be searched using the search feature and a combination of keywords included in online supplemental annex 4; when the search feature is not available, the website will be scanned for relevant information. When the search feature is available, results will be reviewed using the title and short text underneath and until no more relevant items are identified on a following page. All potentially relevant reports will be exported into Covidence and de-duplicated for title and abstract screening.

### Selection

For published literature and grey literature, all title and abstracts (for published literature) or short text (for grey literature) will be screened by two investigators independently using Covidence. Full-texts of all potentially relevant publications will then be acquired and assessed by two investigators independently to establish eligibility for inclusion into the review. Any disagreements in the screening of search results will be resolved by discussion and consultation with a third investigator as needed.

Following the selection process of published and grey literature, field experts—including clinicians, academics, non-governmental staff, the Diabetic Retinopathy Work Group of the International Agency for the Prevention of Blindness and WHO representatives—will be provided with a list of the included reports and requested to nominate further potentially relevant reports. An Excel spreadsheet will be used to track who has been contacted, who has responded, the need for a follow-up message, and identified reports.

### Data collection and data items

Data extraction will be performed in Covidence. The data extraction form will be piloted by two investigators on five studies and modifications undertaken as required. The following details of each report will be extracted by two investigators independently. Any discrepancies will be resolved by discussion, and with a third reviewer if necessary. In the event of unclear outcome data, two attempts will be made to contact the corresponding author by email.

1. Study characteristics: country/countries of study, type of report, source of data, study design, administrative region of study (when subnational eg, province, district, state), year(s) of data collection, number of participants, overall participation rate, extent of missing data.
2. Participant characteristics: inclusion criteria (age, diabetes type), age (median/mean and range), diabetes type, diabetic retinopathy status, vision impairment status,[21] available information across any PROGRESS dimension (outlined above).[17]
3. Outcome(s): definition of screening coverage (numerator, denominator, screening interval), period of follow-up (if applicable), retina screening coverage result (aggregated and if disaggregated by diabetes type, diabetic retinopathy status or any equity/PROGRESS dimensions (outlined above)[17]).

## Outcomes

The primary outcome measure is retina screening coverage for people with diabetes. This is defined in a range of ways. We will include the broad categories of outcomes outlined in Table 1 and will expand this list based on what is identified during the search.

## Risk of bias in individual studies

Two investigators will independently assess the risk of bias in each included study. Any conflict in relation to the appraisal will be discussed between the two investigators and resolved with a third investigator if necessary. Relevant Joanna Briggs Institute critical appraisal tools will be used to evaluate each included study .[22]

## Assessment of heterogeneity

Clinical heterogeneity will be assessed by comparing key participant characteristics at the study level (eg, age, sex, diabetes type, diabetic retinopathy status). Methodological heterogeneity will also be considered, including variation in the specific definition of the numerator and denominator as well as the reporting period and method of data collection. If meta-analysis is possible, we will assess statistical heterogeneity by inspecting forest plots and the $I^2$ and $\tau^2$ statistics to examine the proportion of heterogeneity across studies that is due to chance. If high levels

**Table 1** Anticipated definitions of retina screening coverage for people with diabetes

| Definition category (reference/example) | Details | |
|---|---|---|
| WHO definition and data sources[10] | Data source 1: preferred data source=facility | Definition: the proportion of people with diabetes who undertook a retina examination at the recommended interval.<br>Numerator=number of people with diabetes who undertook a retina examination at the recommended interval.<br>Denominator=total number of people registered in a facility or, if a register is not available, the estimated prevalence of diabetes in the population covered by a facility.<br>Interval=as defined in national screening guidelines or biennially if not outlined in a guideline. |
| | Data source 2: alternative data source=population-based surveys* | Numerator=number of people in the survey who self-report attendance at an eye check for diabetes.<br>Denominator=number of people in the survey with diabetes (either self-reported or confirmed with testing).<br>Interval=either not specified (eg, the participant is asked when they last attended a check) or a specified period (eg, the participant is asked whether they attended a check within the last 12 months/2 years/other time period). |
| Other definitions | Cohort from facility-based data[14] | Numerator=number of people in the cohort who had attended all of the expected retinal screening appointments (as per the national guideline or biennially) during the follow-up period.<br>Denominator=total number of people registered with diabetes OR if that is not available, the number of people with diabetes who had at least one retinal screening appointment during the follow-up period.<br>Interval=as defined in national screening guidelines or biennially if not outlined in a guideline. |
| | Attendance following diabetes diagnosis[27] | Numerator=number of people with newly diagnosed diabetes who attended the first retinal screening appointment.<br>Denominator=total number of people registered with newly diagnosed diabetes.<br>Interval=within a specified period, for example, within 12 months of diabetes diagnosis. |

*The WHO definition requires facility data. For this alternative data source, we have proposed this definition and will reflect on variations of it in reporting of objective 1.

of heterogeneity are detected (eg, I²>50%) and there are sufficient data available (eg, meta-analysis including more than 10 studies), we will explore likely sources of this heterogeneity via meta-regression.

## Data synthesis

We will take an inclusive approach for objective 1 and will then determine the potential for synthesis of results reporting the WHO definition (table 1) for objective 2, before exploring disparities within all included studies in objective 3.

To achieve objective 1, we will outline the full range of definitions used across included studies and summarise the study characteristics of each definition, including geographic (country, national vs subnational), source of data, study design, inclusion criteria and extent of data missingness.

To achieve objective 2, in countries where estimates on retina screening coverage for people with diabetes using the WHO definition are available from multiple sources, we will establish the country estimate according to the decision tree included in online supplemental annex 5. We will first follow the process separately for estimates generated for the preferred and alternative data sources for the WHO definition (facility-based and survey-based, respectively, see table 1). The process outlined in the decision tree prioritises more recent studies and studies with nationally representative sampling frames. Where no nationally representative estimates are available, we will pool two or more subnational estimates if the definitions of the outcome are comparable, and the estimates are close enough in value that the pooled result provides a meaningful measure of retina screening coverage for people with diabetes. We will combine these subnational estimates from the same country using an inverse variance weighted average, using the metagen command from the meta package in R. If two sources of data are then available for a country, we will prioritise facility data in countries with organised retina screening programmes and accurate diabetes registries, and survey data in countries with opportunistic screening or with no centralised registries. We will document the extent to which these judgements affect the final country estimates by a sensitivity analysis.

We will use the country estimates to calculate the median, interquartile range and range of retina screening coverage for each WHO region and World Bank country income level,[23] stratified by screening interval (eg, 1–2 years and 3 years). We will also compare retina screening coverage estimates over time where comparable sampling frames have been used in repeat studies.

To achieve our third objective, and depending on availability of data, we will calculate risk ratios comparing coverage in different groups, specifically:
1. Diabetes type (type 2 vs type 1).
2. Duration of diabetes (per year increase).
3. Diabetic retinopathy status (present vs absent).
and PROGRESS categories:

1. Place of residence (eg, rural vs urban)
2. Ethnicity (eg, non-dominant vs dominant)
3. Occupation status (eg, not-employed vs employed).
4. Gender/sex (eg, female vs male)
5. Religion (eg, non-dominant vs dominant).
6. Education level (eg, no educational qualification vs school level qualification vs further educational qualification).
7. Socioeconomic status (eg, high vs low area level deprivation).
8. Social capital (eg, low vs high social capital).

We will plot these risk ratios in separate forest plots and combine values where appropriate that is, where we judge that pooling will provide a meaningful measure of comparative risk (see the Assessment of heterogeneity section), using the metagen command from the meta package in R. If there are sufficient data available (eg, meta-analysis including 10 or more studies), we will also assess small study effects, one of which may be publication bias, by preparing a funnel plot,[24] which is a scatter plot of effect size versus precision (SE).

## Certainty assessment

Currently there is no formal guidance from GRADE regarding the assessment of certainty in systematic reviews and meta-analysis of prevalence studies.[25] Therefore, we do not plan to assess certainty formally but we will consider the GRADE domains, that is, risk of bias, inconsistency, imprecision, indirectness and publication bias when drawing conclusions from the available evidence.[26]

## Patient and public involvement

Patients and the public will not be involved in the conduct of the systematic review and meta-analysis, as we plan to review existing published literature and publicly available reports only.

## Ethics and dissemination

This review will only include published data thus no ethical approval will be sought. The findings of this review will be published in a peer-reviewed journal and presented at relevant conferences.

**Contributors** JR conceived the idea for the review. NC and JR drafted and revised the protocol with suggestions from PRS, CB, LG, IM, MJB, SK and JE. IG constructed the search.

**Funding** NC is supported by a University of Auckland Doctoral Scholarship. PRS is supported by a Buchanan Charitable Foundation Postdoctoral Fellowship. MJB is supported by the Wellcome Trust (207472/Z/17/Z).

**Disclaimer** The author SK is a staff member of the World Health Organization. The author alone is responsible for the views expressed in this publication and they do not necessarily represent the views, decisions or policies of the World Health Organization.

**Competing interests** None declared.

**Patient and public involvement** Patients and/or the public were not involved in the design, or conduct, or reporting, or dissemination plans of this research.

**Patient consent for publication** Not applicable.

**Provenance and peer review** Not commissioned; externally peer reviewed.

**ORCID iDs**
Nimisha Chabba http://orcid.org/0009-0009-2863-7322
Pushkar Raj Silwal http://orcid.org/0000-0001-6100-090X
Ian McCormick http://orcid.org/0000-0002-7360-3844
Jacqueline Ramke http://orcid.org/0000-0002-5764-1306

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
