## [Reviewer comments · BMJ Open]

ARTICLE DETAILS

TITLE (PROVISIONAL)	What is the coverage of retina screening services for people with diabetes? Protocol for a systematic review and meta-analysis
AUTHORS	Chabba, Nimisha; Silwal, Pushkar; Bascaran, Covadonga; McCormick, Ian; Goodman, Lucy; Gordon, Iris; Burton, Matthew J; Keel, Stuart; Evans, Jennifer; Ramke, Jacqueline

VERSION 1 – REVIEW

REVIEWER	Owusu-Afryjie, Bismark University of Houston College of Optometry
REVIEW RETURNED	08-Nov-2023

GENERAL COMMENTS	Diabetic retinopathy (DR) is a major challenge to public health, and the low uptake of diabetic eye screening services is worrisome. The researchers plan a comprehensive search for literature from databases, reference lists of published articles/reports, grey areas, and consult with field experts for potentially relevant publications on the coverage of DR screening services. The outcome of the proposed study will have national, regional, and global impacts on planning and addressing blindness caused by DR. Below are a few comments for the researchers to address and/or consider: Abstract: Methods and analysis: 1. Line 15: The coverage of databases varies. Therefore, it may be beneficial to include other databases such as ISI Web of Science, Cochrane, etc. While there may be an overlap of articles from the different databases, you may also find unique articles from each database to minimize the risk of omitting relevant articles. Duplicate articles can be removed as you have indicated on page 6, line 14. Otherwise, kindly provide a potential reason for using only Medline and Embase.2. Line 17: Please include the “end date” of the study.3. Line 25: Kindly check the spacing of the words. Strengths and limitations of this study Lines 42 - 43: Please include the “end date” of the study. Methods and analysis: Lines 41-42: Please provide a potential reason for excluding studies reporting during the COVID-19 years. Outcomes: Box 1
--

	Second column line 47: Please check the spacing between "Attendance" and "following." Annex 1: Please re-check the page numbers (location where reported) of items 2, 14, 15a, 15b, and 15c. Annex 3: Some countries are divided into regions, municipalities, and districts instead of states and provinces. In addition, the health sector in some countries is subdivided into directorates of health instead of departments of health. Thus, it may be necessary to include variations such as "municipal directorate of health", "regional directorate of health", etc. Annex 4: Kindly include alternative forms of DR screening such as "diabetic eye exams", "retinal eye exams", "retinal imaging service", "retinal screening", and "diabetic retinopathy service." Thank you.
--	---

REVIEWER	Shah, Payal Sankara Eye Hospital
REVIEW RETURNED	21-Nov-2023

GENERAL COMMENTS	Kindly include the time period/ dates of the study as per the journal guidelines. Please mention specific number, i.e small number of facilities(page 5 line 34) for exclusion criteria. "Small" is a vague term.
--

VERSION 1 – AUTHOR RESPONSE

Reviewer: 1

Dr. Bismark Owusu-Afriyie, University of Houston College of Optometry

Comments to the Author:

Reviewer 1

Diabetic retinopathy (DR) is a major challenge to public health, and the low uptake of diabetic eye screening services is worrisome. The researchers plan a comprehensive search for literature from databases, reference lists of published articles/reports, grey areas, and consult with field experts for potentially relevant publications on the coverage of DR screening services. The outcome of the proposed study will have national, regional, and global impacts on planning and addressing blindness caused by DR. Below are a few comments for the researchers to address and/or consider:

Abstract:

Methods and analysis:

1. Line 15: The coverage of databases varies. Therefore, it may be beneficial to include other databases such as ISI Web of Science, Cochrane, etc. While there may be an overlap of articles from the different databases, you may also find unique articles from each database to minimize the risk of omitting relevant articles. Duplicate articles can be removed as you have indicated on page 6, line 14. Otherwise, kindly provide a potential reason for using only Medline and Embase.

Response: Thank you – in our extensive experience we tend not to find additional studies beyond MEDLINE and Embase. We have, however, expanded the databases included in response to this comment, and have added the searches to our Annex. We made the following additions to the protocol (page 5/line 10):

We will search MEDLINE (Ovid), Cochrane Central Register of Controlled Trials (CENTRAL), Embase, Global Health (Ovid), Emerging Sources Citation Index (ESCI), Science Citation Index Expanded (SCI-EXPANDED), and Social Sciences Citation Index (SSCI) on Web of Science (Clarivate) for reports using data collected since 1 January 2000 using a search strategy developed by an experienced Information Specialist (IG).

We also added the information to the abstract (page 2, line 15):

A search will be conducted in Medline (Ovid) and Embase five databases without language restrictions for studies from any country reporting retina screening coverage for adults with any type of diabetes at the national or subnational level using data collected since 1 January 2000 until the search date.

And made a corresponding change to the Strengths and limitations section (page 2, line 41):

A strength of this review is the comprehensive search, designed by an experienced information specialist, and without any language or geographical restrictions, across two several databases for reports published from 1 January 2000 until the search date.

2. Line 17: Please include the “end date” of the study.

Response: Thank you for this feedback. We wrote the protocol as a ‘forward looking’ document so intended on including the search date in the final review report. We have added ‘until the search date’ to the sentence as shown below. However, if the policy of the journal is to include the date of the search in the protocol, the appropriate date would be 29 September 2023. Our understanding of standard practice in protocols would be to not include this date but leave it to the discretion of the editor. The following addition was made to page 2 line 19:

A search will be conducted in Medline (Ovid), Embase, Cochrane Central Register of Controlled Trials (CENTRAL), Global Health (Ovid), Science Citation Index Expanded, Social Sciences Citation Index and Emerging Sources Citation Index on Web of Science (Clarivate) without language restrictions for studies from any country reporting retina screening coverage for adults with any type of diabetes at the national or subnational level using data collected since 1 January 2000 until the search date.

3. Line 25: Kindly check the spacing of the words.

Response This has now been corrected.

4: Strengths and limitations of this study

Lines 42 - 43: Please include the “end date” of the study.

Response: As outlined for comment 2 above, we have added ‘until the search date’ as this is a protocol and we followed standard practice for the search date to be forward looking (shown below). Should the editor prefer, this can be replaced with the actual search date which was 29 September 2023. The following change was made to page 2, line 45:

A strength of this review is the comprehensive search, designed by an experienced information specialist, and without any language or geographical restrictions, across two databases for reports published from 1 January 2000 until the search date.

5: Methods and analysis:

Lines 41-42: Please provide a potential reason for excluding studies reporting during the COVID-19 years.

Response: We thank the reviewer for this helpful comment. We have now added the following text to explain our reasoning for this to page 4, line 42:

We will exclude studies reporting data exclusively during the peak of the COVID-19 pandemic (2020 and 2021) due to the range of responses and restrictions across the world that reduced health service delivery.

6: Outcomes: Box 1

Second column line 47: Please check the spacing between “Attendance” and “following.”

Response: This has now been corrected.

7: Annex 1:

Please re-check the page numbers (location where reported) of items 2, 14, 15a, 15b, and 15c.

Response: Thank you for this feedback. The location of item 2 has been updated from page 1 to page 2; item 14 from page 6 to pages 7; item 15a from pages 6 to 7 to pages 7 to 8; item 15b from pages 6 to 7 to pages 7 to 8; and item 15c from pages 6 to 7 to page 7.

8: Annex 3:

Some countries are divided into regions, municipalities, and districts instead of states and provinces. In addition, the health sector in some countries is subdivided into directorates of health instead of departments of health. Thus, it may be necessary to include variations such as “municipal directorate of health”, “regional directorate of health”, etc.

Response: Thank you for these suggestions. We agree that there are substantial variations in subnational governance structures and the terms used to refer to these globally. However, while waiting for this peer-review feedback, we have undertaken a pilot of the grey literature search for seven countries, and we have found that most of the search results captured with the search term “health department” are also found with the additional search terms outlined in Annex 3 “provincial health department” and “ministry of health”. Thus, we are confident that the current search strategy will not miss documents that the additional search term variations suggested here would identify. Thus, we have not made any change in response to this comment.

9: Annex 4:

Kindly include alternative forms of DR screening such as “diabetic eye exams”, “retinal eye exams”, “retinal imaging service”, “retinal screening”, and “diabetic retinopathy service.”

Response: Thank you for these suggestions. As outlined in comment 8 above, we have completed a pilot of the grey literature search for seven countries, and we have found that most of the search results captured with the search term “diabetic eye screening” are captured with the search term “diabetic retinopathy screening.” Similarly, most of the search results found with the search term “retina screening coverage” are also found with the search term “retina screening AND adherence”. Thus, we are confident that the current search strategy will cover the suggested search term variations.

Reviewer: 2

Dr. Payal Shah, Sankara Eye Hospital

Comments to the Author:

1: Kindly include the time period/ dates of the study as per the journal guidelines.

Response: Thank you for this feedback. In terms of completing the review, this review will be completed within the scope of the first author’s PhD research. However, if the reviewer is referring to the search dates, please see our response to reviewer 1, comment 2. Should the editor prefer, we are open to including an approximate completion date of the review – the conservative date we included in the protocol registration was 1 June 2025.

2: Please mention specific number, i.e small number of facilities (page 5 line 34) for exclusion criteria. “Small” is a vague term.

Response: We thank the reviewer for this helpful comment. We have now updated the text to remove this ambiguity (page 4, line 34):

We will exclude studies reporting data where from a small number of facilities unless we consider the sampling is not representative of the national or state/province/district level (e.g. in small island states).